# Anisotropic Message Passing: Graph Neural Networks with Directional and Long-Range Interactions

**Moritz Thürlemann, Sereina Riniker**
Department of Chemistry and Applied Biosciences
ETH Zürich
Vladimir-Prelog-Weg 2, 8093 Zürich, Switzerland
`moritzt@ethz.ch, sriniker@ethz.ch`

## Abstract

Graph neural networks have shown great potential for the description of a variety of chemical systems. However, standard message passing does not explicitly account for long-range and directional interactions, for instance due to electrostatics. In this work, an anisotropic state based on Cartesian multipoles is proposed as an addition to the existing hidden features. With the anisotropic state, message passing can be modified to explicitly account for directional interactions. Compared to existing models, this modification results in relatively little additional computational cost. Most importantly, the proposed formalism offers as a distinct advantage the seamless integration of (1) anisotropic long-range interactions, (2) interactions with surrounding fields and particles that are not part of the graph, and (3) the fast multipole method. As an exemplary use case, the application to quantum mechanics/molecular mechanics (QM/MM) systems is demonstrated.

## 1 Introduction

Message passing graph neural networks (GNN) have shown great potential for the description of a wide range of chemical systems (Scarselli et al., 2009; Battaglia et al., 2016; 2018). Particularly the description of quantum molecular (QM) systems with machine learning (ML) potentials has received a lot of interest (Gilmer et al., 2017; Unke et al., 2021b). However, in its general form, message passing does not explicitly account for directionality, which plays an important role in many physical interactions (Glotzer & Solomon, 2007; Kramer et al., 2014). In recent years, a range of models which include directional information have been proposed (Anderson et al., 2019; Klicpera et al., 2020; Miller et al., 2020; Schütt et al., 2021). Especially models based on Clebsch-Gordan tensor products have shown superior data efficiency (Batzner et al., 2022; Batatia et al., 2022b; Musaelian et al., 2022). However, the relatively high cost of these operations might hinder the application to larger systems such as biomolecules in solution. This difficulty is compounded by growing evidence that message passing cannot accurately resolve long-range interactions (Alon & Yahav, 2020; Dwivedi et al., 2022). Note that we consider here interactions as long-range if convergence in real space is slow or non-existent, e.g. electrostatic interactions or polarization.

In light of these challenges, which are exemplary illustrated in Figure 1, we propose a model with the aim to (1) include directional information while (2) retaining computational efficiency and (3) incorporating (anisotropic) long-range interactions. Specifically, the addition of an anisotropic state to the existing hidden features is proposed. This anisotropic state is based on Cartesian multipoles and expressed as a linear combination of local frames. As a result, these multipoles are equivariant under rotations and can be used to describe anisotropy of interactions. The formalism is developed analogously to the concept of atomic multipoles commonly used in computational chemistry (Stone, 2013). The proposed modification results in relatively little computational overhead compared to standard message passing models. Most importantly, the formulation based on multipoles allows for the hybrid treatment of particles and fields. This is of particular interest for two cases: (1) Systems of particles embedded in an external field, and (2) systems with large numbers of particles where long-range interactions may be treated with the fast multipole method (Rokhlin, 1985).

As a practical use case, we consider quantum mechanics/molecular mechanics (QM/MM) simulations (Warshel & Levitt, 1976). In QM/MM simulations, a QM system is embedded in an external electrostatic field generated by MM particles, allowing for an efficient description of large systems, for instance protein-ligand complexes or enzymatic reactions in solution. As such, the QM/MM formalism might be ideally suited to apply ML potentials to extended systems. Consequently, a ML/MM formalism within the message passing framework is formulated analogous to QM/MM.

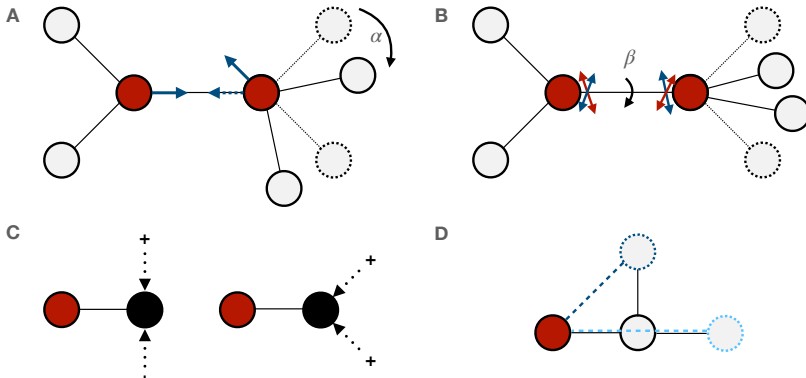

Figure 1: Exemplary situations arising in molecular systems for which message passing with distance labelled edges fails. (**A**): Rotation of one molecule by an angle $\alpha$ cannot be recognized without adding more edges. However, this change can be resolved through the addition of a dipole (blue arrow). (**B**): Similarly, a rotation by an angle $\beta$ around the axis described by the aligned dipoles ('torsion') can be resolved by including a quadrupole (blue/red crossed arrows). (**C**): An electric field caused by charges (+) surrounding the molecule. While the electrostatic interaction is additive, the polarization on the black atom due to the opposing charges cancels out in the left case but adds up in the right case. (**D**): 'Long-range' interactions (dashed lines): Message passing can neither distinguish distance nor direction for the interactions between the red and the two blue-dotted particles.

## 2 RELATED WORK

ML potentials have sparked interest as a possible solution to the steeply scaling computational cost of quantum chemical methods (Schuch & Verstraete, 2009; Unke et al., 2021b), with high-dimensional neural network potentials being some of the earliest proposed methods (Behler, 2011). With the introduction of GNN (Scarselli et al., 2009; Battaglia et al., 2016; 2018), the development of ML potentials has shifted away from handcrafted features to end-to-end differentiable models which learn features solely from distances and element types (Gilmer et al., 2017; Unke et al., 2021b). Recognizing that GNN may fail to distinguish certain graphs has spurred the development of modified message passing schemes (Morris et al., 2019; Maron et al., 2019; Pozdnyakov & Ceriotti, 2022). As a solution, the explicit integration of many-body interactions in the message passing (Kondor, 2018; Klicpera et al., 2020; Shui & Karypis, 2020; Zhao et al., 2021), directional messages (Anderson et al., 2019; Schütt et al., 2021), and combinations thereof (Gasteiger et al., 2021; Batatia et al., 2022b) have been proposed. Building on the concept of equivariant CNNs (Cohen & Welling, 2016; Weiler et al., 2018), equivariant message passing models based on Clebsch-Gordan tensor products have received increased attention (Anderson et al., 2019; Miller et al., 2020; Satorras et al., 2021; Brandstetter et al., 2021; Batzner et al., 2022; Musaelian et al., 2022). A categorization based on employed architectural features was recently proposed by Batatia et al. (2022a), offering a succinct overview. Deficiencies in the description of long-range interactions have led to models that include explicit interaction terms (Unke & Meuwly, 2019; Ko et al., 2021), as well as models that integrate this information in the representation itself (Grisafi & Ceriotti, 2019; Grisafi et al., 2021; Unke et al., 2021a). As an alternative to the ML potentials mentioned so far, ML within the QM/MM formalism might facilitate the description of condensed phase systems. Recent work has highlighted the challenges and potential of ML for the simulation of QM/MM systems (Zhang et al., 2018; Böselt et al., 2021; Pan et al., 2021; Hofstetter et al., 2022; Giese et al., 2022).

## 3 METHODS

### 3.1 MESSAGE PASSING GRAPH NEURAL NETWORKS

Message passing GNN process graph-structured data. In its commonly used form, node, edge, and/or global features are iteratively refined. Various models have been proposed in recent years with differences found in the construction of the graph and the employed message passing mechanism (Scarselli et al., 2009; Battaglia et al., 2016; Gilmer et al., 2017; Battaglia et al., 2018).

Considering a graph $\mathcal{G} = (\mathcal{V}, \mathcal{E})$ with nodes $\boldsymbol{v}_i \in \mathcal{V}$ and edges $\boldsymbol{e}_{ij} \in \mathcal{E}$, message passing can be defined as

$$\boldsymbol{h}_i^{l+1} = \phi_h(\boldsymbol{h}_i^l, \sum_{j \in N(i)} \phi_e(\boldsymbol{h}_i^l, \boldsymbol{h}_j^l, \boldsymbol{u}_{ij})) \tag{1}$$

with $\boldsymbol{h}_i^l \in \mathbb{R}^n$ describing the hidden-feature vector of node $\boldsymbol{v}_i$ after $l$ iterations, $\boldsymbol{u}_{ij} \in \mathbb{R}^n$ the edge feature of the edge $\boldsymbol{e}_{ij}$ between node $i$ and $j$, $N(i)$ denoting the set of neighbours of $\boldsymbol{v}_i$, and $\phi_e$ and $\phi_h$ describing edge and node update functions. The superscript $l$ denotes the current message passing iteration with $n$ being the total number of message passing layers.

In the following paragraphs, we will refer to QM and MM particles. In the context of this work, QM particles are nodes $\boldsymbol{v}_i$ of a graph $\mathcal{G}$, which are labelled with a hidden feature $\boldsymbol{h}_i$. MM particles, on the other hand, are not part of the graph, and therefore do not hold a hidden feature. Instead, only a scalar partial charge $q_i$ (monopole) is attached to MM particles. More generally, MM particles can be understood as a specific example of a source of an external field in which the graph is embedded. In addition, both QM and MM particles hold a position $\boldsymbol{r}_i \in \mathbb{R}^3$ in Cartesian space.

### 3.2 THE ANISOTROPIC STATE

The present work proposes the introduction of an internal anisotropic state $\mathbf{M}_i$ composed of Cartesian multipoles of order $k$ to the existing hidden features. Cartesian multipoles are ideally suited for this task as they result in relatively little additional computational costs and conceptual complexity. Specifically, $\mathbf{M}_i^k \in \mathbb{R}^{3^n}$ refers to the $k$-th moment of the anisotropic state of node $i$ composed of the monopole $\mathbf{M}_i^0$, the dipole $\mathbf{M}_i^1$ and so on. For clarity, the iteration superscript $l$ will be omitted. However, in practice, the anisotropic state can be updated during each iteration much like the hidden feature $\boldsymbol{h}_i^l$.

The anisotropic state is expressed as a linear combination of a local basis (Thürlemann et al., 2022). This formulation preserves equivariance under rotations and invariance to translations. As the local basis, the traceless tensor product of order $k$ of the unit vector $\hat{\boldsymbol{r}}_{ij}$ pointing from the position of node $i$ to the position of its neighbour $j$ is used

$$\mathbf{R}_{ij}^k = \hat{\boldsymbol{r}}_{ij} \otimes \hat{\boldsymbol{r}}_{ij} \otimes ... \tag{2}$$

with $k$ referring to the order of the multipole and $\mathbf{R}_{ij}^0 = 1$. Each component of order $k$ of the anisotropic state $\mathbf{M}_i^k$ is then expressed as

$$\mathbf{M}_i^k = \sum_{j \in N(i)} c_{ij}^k \mathbf{R}_{ij}^k \tag{3}$$

with coefficients $c_{ij}^k$ predicted for each interaction as

$$c_{ij}^k = \phi_{M(k)}(\boldsymbol{h}_i, \boldsymbol{h}_j, \boldsymbol{u}_{ij}) \tag{4}$$

where $\phi_{M(k)}$ refers to a learnable function. The scalar coefficient $c_{ij}^k$ is based on the hidden features of the interacting nodes $i$ and $j$ as well as an edge feature $\boldsymbol{u}_{ij}$. The coefficient $c_{ij}^k$ can be interpreted as the contribution of the node $j$ to the polarization of order $k$ of node $i$ analogous to a (hyper-)polarizability. We note that the multipoles discussed here serve as a tool to introduce anisotropy. Albeit possible, they do not necessarily carry physical meaning.

### 3.3 ANISOTROPIC INTERACTION FEATURES

Given the anisotropic state $\mathbf{M}_i$ proposed in the previous section, multipole-multipole interactions can be defined. Formulation of the multipole interaction follows the notation and formalism described in (Smith, 1998; Lin, 2015; Burnham & English, 2020) which decomposes the interaction between multipoles on two distinct sites into a product of a radial component $b_k(||r_{ij}||)$ and coefficients of the multipole interactions $g_{ij}^k(\boldsymbol{r}_{ij})$, resulting in the following terms

$$u_{ij} = \sum_{k=0}^{\infty} b_k(||\boldsymbol{r}_{ij}||)g_{ij}^k(\boldsymbol{r}_{ij}) \tag{5}$$

with $b_k(||\boldsymbol{r}_{ij}||) = (2k-1)!!/||\boldsymbol{r}_{ij}||^{2k+1}$. In the context of machine learning, $b_k$ can be reinterpreted as filters analogous to the continuous-filter convolutions introduced by Schütt et al. (2018). Consequently, the coefficients $g_{ij}^k$ encode the angular information based on the relative orientation of the multipoles centered on the interacting nodes. In the notation of Burnham & English (2020), the coefficients can be generated with

$$\boldsymbol{g}_{ij}^k = \bigoplus_{d_i+d_c+d_j=k} \langle (\mathbf{M}_i^{(d_i+d_c)}.d_i.\mathbf{R}_{ij}^{d_i}), (\mathbf{M}_j^{(d_j+d_c)}.d_j.\mathbf{R}_{ij}^{d_j}) \rangle \tag{6}$$

given multipoles $\mathbf{M}_i^{(d_i+d_c)}$ of order $(d_j + d_c)$ on site $i$ and the previously defined traceless tensor products of the vector $\hat{\boldsymbol{r}}_{ij}$. $\bigoplus$ refers to the concatenation of the scalar coefficients. All $d_i, d_j, d_c$ where $d_i+d_j+d_c = k$ and $d_i, d_j, d_c \geq 0$ are included. $d$ denotes the number of contractions, with $d_i$ being the number of contractions over the indices of the first bracket, $d_j$ the number of contractions in the second bracket and $d_c$ the number of contractions between the two brackets, indicated by $\langle .,.\rangle$. Analogous to the multipole interaction defined in Eq. 5, the anisotropic feature $\boldsymbol{a}_{ij}$ is defined as

$$\boldsymbol{a}_{ij} = \phi_b(||\boldsymbol{r}_{ij}||) \circ G(\hat{\boldsymbol{r}}_{ij}, \mathbf{M}_i, \mathbf{M}_j) \tag{7}$$

where we use $\circ$ to denote the aforementioned element-wise multiplication with radial weights generated by $\phi_b$ and $G(\hat{\boldsymbol{r}}_{ij}, \mathbf{M}_i, \mathbf{M}_j)$ for the concatenated multipole interaction coefficients $\boldsymbol{g}_{ij}^k$ of all considered orders $k$.

### 3.4 ANISOTROPIC MESSAGE PASSING (AMP)

Based on the anisotropic feature defined in the previous section, the modified message passing is obtained as

$$\mathbf{M}_i^k = \sum_{j \in N(i)} \phi_{M(k)}(\boldsymbol{h}_i^l, \boldsymbol{h}_j^l, \boldsymbol{a}_{ij})\mathbf{R}_{ij}^k$$
$$\boldsymbol{a}_{ij} = \phi_b(||\boldsymbol{r}_{ij}||) \circ G_{ij}(\hat{\boldsymbol{r}}_{ij}, \mathbf{M}_i, \mathbf{M}_j) \tag{8}$$
$$\boldsymbol{h}_i^{l+1} = \phi_h(\boldsymbol{h}_i^l, \sum_{j \in N(i)} \phi_e(\boldsymbol{h}_i^l, \boldsymbol{h}_j^l, \boldsymbol{a}_{ij}))$$

Thus, two additional steps are added to the standard message passing formalism: (1) Expansion of the multipoles on each node, and (2) contraction of the multipoles for each pair of interacting nodes. The resulting feature is used to incorporate directional information in the message $\boldsymbol{m}_{ij}$. For the use as a ML potential, the total potential energy is expressed as a sum of atom-based contributions of each QM particle, i.e. $V_{QM} = \sum_i^{QM} \phi_V(\boldsymbol{h}_i^n)$. We will refer to this model as AMP($k$) where $k$ denotes the highest order of the multipoles used and $n$ is the number of graph layers.

### 3.5 QUANTUM MECHANICS/MOLECULAR MECHANICS (QM/MM)

QM/MM has become an integral part of computational chemistry since its inception in 1976 (Warshel & Levitt, 1976). QM/MM offers an efficient way to incorporate interactions with an environment, for instance solvent molecules and/or proteins, through combination of a QM and a classical (MM) description of a system. As such it retains the fidelity of QM-based methods for the region of interest, e.g. the solute, and the computational efficiency of classical force fields for the surroundings. Here, a brief description of the electrostatic embedding, one of several possible QM/MM schemes, is given. We refer the interested reader to Senn & Thiel (2007) for further details.

Within an electrostatic embedding, a designated QM zone interacts with the partial charges of the surrounding MM particles. For this purpose, the molecular Hamiltonian $\hat{H}_{QM}$ is extended by an interaction with the MM zone

$$\hat{H}_{QM/MM} = -\sum_i^{MM}\sum_j^{EL} \frac{q_i}{||\boldsymbol{r}_i - \boldsymbol{r}_j||} + \sum_i^{QM}\sum_j^{MM} \frac{Z_i q_j}{||\boldsymbol{r}_i - \boldsymbol{r}_j||} \tag{9}$$

where $q_i$ are the partial charges of the MM zone and $Z_i$ refer to the core charges of the nuclei. $MM$ refers to the MM particles, $EL$ to the electrons in the QM zone and $QM$ to the QM nuclei. While the first term incorporates the interaction of the surrounding partial charges analogous to the core-electron interaction, the second term describes the Coulomb interaction between the nuclei within the QM zone and the partial charges. In other words, the QM system interacts with point charges representing the MM particles. The point charges polarize the QM system and interact with the resulting charge distribution. Remaining interactions between QM and MM particles, i.e. exchange-repulsion and dispersion, are described with classical force-field terms such as the Lennard-Jones potential (Jones & Chapman, 1924).

### 3.6    QM/MM for Message Passing: Direct Polarization

As a synthesis of the two previous sections, we propose a further modification to anisotropic message passing, which incorporates interactions with a surrounding field or particles that are not part of the graph.

$$\mathbf{M}_i^k = \sum_{j \in N_{QM}(i)} \phi_{M_{QM}(k)}(\boldsymbol{h}_i^l, \boldsymbol{h}_j^l, \boldsymbol{a}_{ij})\mathbf{R}_{ij}^k + \sum_{j \in N_{MM}(i)} \phi_{M_{MM}(k)}(\boldsymbol{h}_i^l, \boldsymbol{a}_{ij})\mathbf{R}_{ij}^k \tag{10}$$

Here, $N_{QM}(i)$ refers to the neighbours of $i$ in the QM zone and $N_{MM}(i)$ to the neighbours of $i$ in the MM zone. In addition to the previously described polarization due to neighbouring atoms that are part of the graph, a second term is introduced, which adds a contribution caused by the MM particles. The second term is conditioned on the hidden feature of the QM particle and the anisotropic feature $\boldsymbol{a}_{ij}$, which is based on the multipoles on the QM particle $i$ and the partial charge on the MM particle. Evidently, this formulation permits also the application to polarizable embeddings schemes, use with the fast multipole method, and use of higher-order multipoles. We note that MM particles contribute only to the polarization of multipoles with order $k \geq 1$, analogous to the definition of (hyper-)polarizabilities (Stone, 2013). Since MM particles are only labelled with a charge and not with a hidden feature $\boldsymbol{h}_i^l$, the interaction depends solely on the feature of the QM particle and the anisotropic interaction feature $\boldsymbol{a}_{ij}$, which encodes the strength of the interaction and the relative orientation of the MM particle with respect to the polarized QM particle. The advantages of the proposed interaction mechanism are twofold: First, the evaluation of messages between MM particles and between QM and MM particles is avoided. Instead, only the cheaper polarization term is evaluated for the more numerous QM–MM interactions. Second, directional and long-range interactions are efficiently incorporated in addition to the proper cancellation of opposing polarization terms. The total potential energy is then obtained as the atom-based contribution due to the QM particles described previously and a contribution due to the Coulomb interaction between QM and MM particles,

$$V_{QM/MM,ESP} = \sum_i^{QM}\sum_j^{MM} \frac{\phi_\rho(\boldsymbol{h}_i^n) \cdot q_j}{||\boldsymbol{r}_{ij}||} \tag{11}$$

where $q_j$ denotes the partial charge of the respective MM particle and $\phi_\rho(\boldsymbol{h}_i^n)$ refers to a scalar charge density localized on the respective QM particle, analogous to a discretized Coulomb interaction. In other words, a pairwise interaction between each QM atom and its neighbours in the MM zone is added. In principle, $\phi_\rho(\boldsymbol{h}_i^n, \boldsymbol{a}_{ij})$ may be used to incorporate anisotropy. Here, only the hidden feature $\boldsymbol{h}_i^n$ is used. Thus, asymptotically correct long-range behaviour can be guaranteed by enforcing that the resulting scalar charges conserve total charge within the QM zone by subtraction of the mean excess charge, i.e. $\sum_i^{QM} \phi_\rho(\boldsymbol{h}_i^n) = 0$ for neutral systems. Expansion of $\phi_\rho(\boldsymbol{h}_i^n)$ itself may be an alternative route to obtain anisotropic QM/MM electrostatic interactions, as shown by Grisafi & Ceriotti (2019). The forces $F_i$ acting on the QM and MM particles are obtained as the derivative of the negative total energy $(V_{QM/MM,ESP} + V_{QM})$ with respect to the positions $\boldsymbol{r}_i$ of the QM and MM particles, respectively. We will refer to models that include this direct polarization terms as AMP($k$)-D. Use of the pairwise long-range potential in Eq. 11 will be denoted by '-P'.

## 4 RESULTS

Results are split into three parts: First, two model systems are investigated to explore the proposed modifications. Second, the model is applied to an existing dataset of QM/MM systems. Finally, results for QM9 are reported as a comparison with existing models.

### 4.1 WATER DIMERS: DIRECTIONAL FEATURES

As an illustrative example of the discussed concepts, a water dimer as shown in Figure 1B is investigated. Configurations were generated by rotating one molecule in steps of one degree around the axis described by the aligned dipoles, resulting in a total of 360 configurations. The molecules were separated by a distance of 4 Å. For such a system, three cutoff regimes are significant: (1) The cutoff $r_{cut}$ used to construct the graph is larger than the maximal distance $r_{max}$ in the system, resulting in a fully-connected graph ($r_{max} < r_{cut}$). (2) The cutoff is larger than the distance $r_{OO}$ between the two opposing oxygen atoms but smaller than the maximal distance ($r_{OO} < r_{cut} < r_{max}$). (3) The cutoff is smaller than the distance between the oxygen atoms, resulting in two disconnected graphs ($r_{cut} < r_{OO}$). Particularly case (2) is interesting as it can be related to torsional degrees of freedom, which play an important role in molecular dynamics.

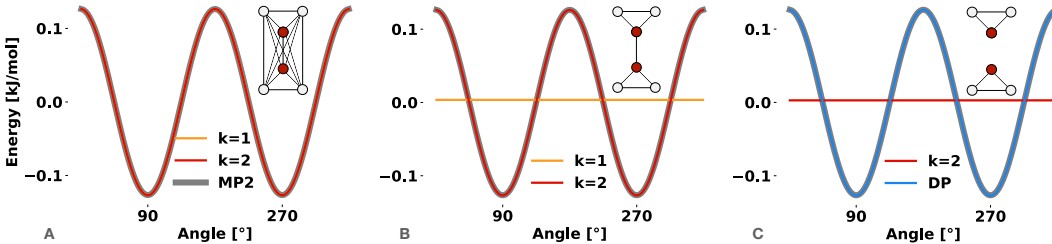

Figure 2: Potential energy ($y$-axis) for the quadrupole-quadrupole interaction between two water molecules for the proposed cutoff regimes with $r_{cut}$ = 8.0 Å (A), $r_{cut}$ = 4.25 Å (B), and $r_{cut}$ = 2.0 Å (C). The relative orientation is indicated by the angle on the $x$-axis. The resulting graph topology for each cutoff is shown as inset. The reference (MP2) is shown in grey. Results for AMP(1), AMP(2), and AMP(2)-DP are shown as orange, red, and blue lines, respectively.

For all results shown in this section, a minimal 1-layer model was used. While the $k = 1$ model is still able to describe the interaction using a cutoff that results in a fully connected graph (i.e. case (1) shown in Figure 2A), a marginally larger error MAE (0.32 AMP(1) versus 0.05 J mol$^{-1}$ AMP(2)) was found for the AMP(1) compared to the AMP(2) model. This observation could be an indication that the present task is also more challenging for AMP(1) than AMP(2). Learning curves shown in Figure 5 add further evidence to this interpretation. For case (2) shown in Figure 2B, the $k = 1$ model (orange line) fails to describe the interaction, as the dipole-dipole interaction remains invariant under the considered rotation. This issue can be traced back to the quadrupole–quadrupole interaction of the opposing oxygen atoms, motivating the need for $k \geq 2$. We note that this case is also interesting as it exemplary demonstrates the limitations of two existing directional message passing architectures, i.e. DimeNet and PaiNN Klicpera et al. (2020); Schütt et al. (2021). These two models seem to fail to describe this quadrupole-quadrupole interaction despite the use of directional information. Our interpretation is that for DimeNet this is due to the invariance of bond angles under the considered rotation, and for PaiNN because it only uses vectorial information (i.e. $k = 1$). Finally for case (3) shown in Figure 2C, all message passing models without additional long-range interactions fail (red line), motivating the proposed pairwise interaction and contributions to the polarization (models with suffix '-D' and '-P', shown in blue). We note that the long-range interaction would in principle also include dispersion, which we do not address here due to the availability of robust and efficient classical models.

### 4.2 WATER CLUSTERS: EXTENSIVITY

For this task, clusters of water molecules, ranging in size from six to 20 molecules, were investigated. For each cluster, one molecule was randomly selected as the QM zone while all other molecules

were treated as MM particles. The goal of this task is the prediction of the difference in energy and gradients caused by placing the QM system in the external field generated by the MM particles. A successful model must be able to discriminate between (1) the polarization of the QM zone by the external field, which does not directly scale with cluster size, and (2) the electrostatic interaction, which increases with cluster size. All models were trained on $1'289$ samples from clusters of size 10. All other cluster sizes were used for testing. Results are shown in Figure 3.

*Baseline – Message Passing:* As a baseline message passing model, the PaiNN architecture is used. PaiNN was shown to perform accurately on a wide variety of tasks through inclusion of directional information (Schütt et al., 2021). For our QM/MM task, the MM particles are part of the same graph as the QM particles. However, energies are only predicted for the three QM particles of the water molecule in the QM zone.

*Ablation Studies – Multipole Order and Polarization:* In addition, the role of the multipole order $k$ was investigated. To compensate for different sizes of $\boldsymbol{a}_{ij}$ due to the multipole order, the number of radial weights was adjusted such that the total number of trainable parameters ($\sim$ 1M) remained constant.

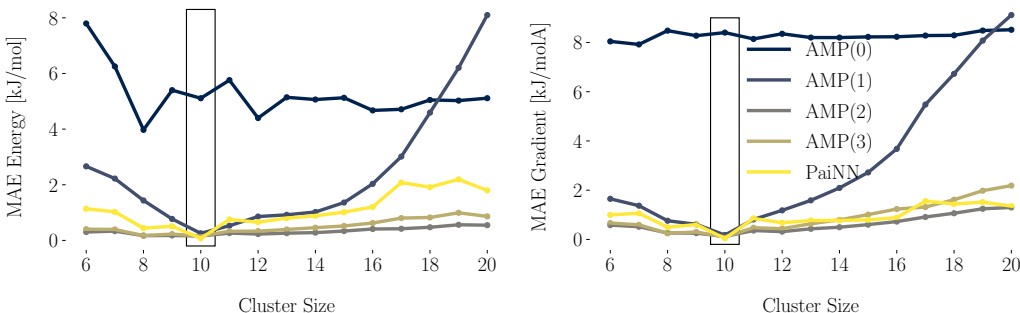

Figure 3: Mean absolute errors (MAE) with respect to the reference (MP2) for the energy ($E_{QM}$) and gradients ($F_{QM}$) of QM/MM water clusters of different sizes. In all cases, one water was treated as a QM system within the electric field of the surrounding charges. All models were trained only on clusters of size 10. Energies are reported in kJ mol$^{-1}$ and forces in kJ mol$^{-1}$ Å$^{-1}$.

Not surprisingly, AMP(0) fails at this task as it only interacts with the MM particles through the pairwise Coulomb potential of Eq. 11. The results for AMP(0) provide a rough estimate of the role of both polarization and anisotropy in the present task. Including the direct polarization (i.e., AMP(0) versus AMP(1)), results in the largest relative improvement. This result demonstrates that the proposed modification allows the model to incorporate information about the surrounding particles without explicit message passing between the QM and MM zone. Interestingly, increasing $k$ from 2 to 3 (i.e., AMP(2) versus AMP(3)), seems to diminish the extrapolation capabilities again, most notably on the largest clusters. Overall, we find that all models that interact with the MM particles (i.e., AMP(1), AMP(2), AMP(3) and PaiNN), perform comparably on the training set and clusters with similar size, whereas differences in performance arise for much larger and smaller clusters. While the performance of the PaiNN model is comparable to the AMP(2) and AMP(3) models on gradients, larger deviations are observed for energies. This result could be an indication that message passing models like PaiNN may have difficulties to differentiate between the electrostatic and polarization interaction in the present task. Building models that differentiate between interaction types might be an important step for ML potentials to improve transferability to large and/or condensed-phase systems.

## 4.3 QM/MM SYSTEMS

As a more realistic example, the model was tested on a previously published datasets of QM/MM systems introduced by Böselt et al. (2021). These datasets include QM energies, QM gradients, and MM gradients of five molecules of varying size solvated in water. In this work, we focus on the three largest and most challenging systems: uracil, retinoic acid (RA) and the S-adenosyl methionine/cysteine dimer (SAM) with 12, 50 and 49 + 14 QM atoms, respectively. On average,

each molecule is surrounded by approximately $1'491$ (uracil), $2'391$ (RA) and $2'553$ (SAM) MM particles. Each water molecule is represented by three partial charges centered on each atom. These datasets are challenging due to the large number of surrounding MM particles that polarize the QM system, the resulting electrostatic long-range interactions, as well as the size and flexibility of RA and SAM. A $\Delta$-ML approach, which learned a correction on top of a semi-empirical tight-binding method (DFTB (Elstner et al., 1998)), was used in previous work (Böselt et al., 2021; Hofstetter et al., 2022).

Table 1: MAE for the QM energy ($E_{QM}$), QM gradients ($F_{QM}$), and MM gradients ($F_{MM}$) of the three QM/MM systems uracil, RA and SAM in water as described in Böselt et al. (2021). Energies are reported in kJ mol$^{-1}$ and forces in kJ mol$^{-1}$ Å$^{-1}$. The terms "unshuffled" and "shuffled" refer to the dataset with and without temporal ordering, respectively.

| System | AMP(3)-DP 50 Å (unshuffled) | | | AMP(3)-DP 50 Å (shuffled) | | |
|---|---|---|---|---|---|---|
| | $E_{QM}$ | $F_{QM}$ | $F_{MM}$ | $E_{QM}$ | $F_{QM}$ | $F_{MM}$ |
| Uracil | 0.2/3.1/2.5 | 0.4/7.3/7.1 | 0.04/0.08/0.09 | 0.3/0.3/0.3 | 0.6/0.6/0.6 | 0.03/0.03/0.03 |
| RA | 0.5/9.1/12.0 | 1.2/5.0/9.6 | 0.06/0.16/0.19 | 0.6/0.6/0.6 | 1.4/1.5/1.5 | 0.06/0.06/0.06 |
| SAM | 2.2/29.1/44.3 | 3.3/16.1/16.0 | 0.20/0.39/0.34 | 3.1/3.2/3.3 | 3.2/3.5/3.5 | 0.15/0.15/0.15 |

No $\Delta$-learning scheme was used in this work. In Table 1, results for a 3-layer AMP(3)-DP model are reported. The performance is compared for two pre-processing steps: While (unshuffled) uses the temporally sequential ordering used in the initial work Böselt et al. (2021); Hofstetter et al. (2022), (shuffled) refers to a shuffled (i.e. randomized) version of the same data. Comparing the same dataset (unshuffled), we observe in most cases larger errors compared to the previously reported $\Delta$-models but smaller errors than previously reported results without $\Delta$-learning. For uracil, (unshuffled) errors were reported without $\Delta$-learning: Using a HDNNP architecture, Böselt et al. (2021) reported MAEs of $2.9/6.6/4.8$ kJ mol$^{-1}$ for $E_{QM}$, $6.38/12.6/12.4$ kJ mol$^{-1}$ Å$^{-1}$ for $F_{QM}$ and $0.62/0.58/0.65$ kJ mol$^{-1}$ Å$^{-1}$ for $F_{MM}$ respectively. With a GNN architecture, Hofstetter et al. (2022) reported MAEs of $7.3$ kJ mol$^{-1}$ for $E_{QM}$, $13.52$ kJ mol$^{-1}$ Å$^{-1}$ for $F_{QM}$ and $0.58$ kJ mol$^{-1}$ Å$^{-1}$ for $F_{MM}$ on the test-set. These results suggest that the proposed model can successfully incorporate long-range polarization and electrostatic interactions. However, the relatively large drop in accuracy for the (unshuffled) dataset, which was previously observed and discussed in Hofstetter et al. (2022), indicates a limited capability to generalise. The difference between random splits or time splits might be important for commonly used benchmark datasets such as MD17 (Chmiela et al., 2017), as random splits tend to give a too optimistic performance assessment. This issue is already well known in biological activity prediction Sheridan (2013). Calculating QM energy, QM gradients, and MM gradients for uracil (12 QM atoms and an average of 1490 MM atoms) using a 3-layer AMP(3)-DP takes around 10 ms on a Nvidia Titan V. Additional results regarding computational costs are reported in the Appendix A.1. Thus, a (QM)ML/MM simulation based on the AMP model could in principle remove the bottleneck of current QM/MM molecular dynamics (MD) simulations almost entirely. However, further improvements are required to bring the cost down to a level comparable to the calculation of classical force fields, which is on the order of $1$ ms/step for a system with around $14'000$ atoms (Eastman et al., 2017). Such improvements are clearly within reach, for instance through the use of the fast multipole method (Rokhlin, 1985). Nevertheless, it is important to mention that a low MAE does not guarantee a stable MD simulation (Stocker et al., 2022). In practice, a $\Delta$-learning scheme might still be required to obtain stable simulations over long timescales. The robustness of the AMP($k$)-DP model for the propagation of (QM)ML/MM MD simulations will therefore be investigated in the future.

Table 2: Comparison of long-range cutoffs. Energies are reported in kJ mol$^{-1}$ and forces in kJ mol$^{-1}$ Å$^{-1}$. The models used here were trained for fewer epochs than those in Table 1, resulting in slightly larger errors. The shuffled dataset was used. Results are given for a training/test split ($7'000/3'000$ data points).

| System | AMP(3)-DP 30 Å | | | AMP(3)-DP 50 Å | | |
|---|---|---|---|---|---|---|
| | $E_{QM}$ | $F_{QM}$ | $F_{MM}$ | $E_{QM}$ | $F_{QM}$ | $F_{MM}$ |
| Uracil | 0.3/0.4 | 0.8/0.9 | 0.06/0.06 | 0.3/0.3 | 0.8/0.9 | 0.03/0.03 |
| RA | 5.3/5.5 | 2.9/3.1 | 0.14/0.14 | 1.5/1.6 | 2.5/2.7 | 0.13/0.13 |
| SAM | 6.0/6.4 | 6.1/6.4 | 0.45/0.45 | 5.5/5.7 | 5.3/5.5 | 0.33/0.33 |

In addition, two long-range cutoffs, $30$ Å and $50$ Å, for the interactions between QM and MM particles are investigated (Table 2). Notable performance differences are observed for the two investigated long-range cutoffs, which reaffirms the importance of long-range interactions. In both cases, a smooth cutoff function is used, thus the differences cannot simply be explained by cutoff artifacts. In this context, it is further important to mention that an increased long-range cutoff does not change the number of parameters and QM–QM interactions but only the number of QM–MM interactions. A $50$ Å cutoff covers all observed QM–MM interactions for all systems. For a $30$ Å cutoff, the same is only true for uracil. Particularly for retinoic acid, for which the solvent-shell is elongated, the difference of the MAE for $E_{QM}$ between the two cutoffs is considerable. Hence, even if a message passing architecture with six layers and a cutoff of $5$ Å, could accurately resolve long-range interactions, these interactions would still fall outside the perceptive radius. However, given that there is little evidence that message passing models can faithfully resolve and incorporate such long-range interactions in the first place, this observation might be crucial for future work on ML potentials.

## 4.4 QM9 DATASET

Results for a $4$-layer AMP(3) model with a cutoff of $5$ Å for the 12 properties of the QM9 dataset are given in Table 3 and compared with existing models reported in the literature. Inference of a $4$-layer AMP(3) model used here requires approximately $26$ ms for a batch of 100 samples with a total of $1'833$ atoms (see Appendix A.1, NVIDIA Titan V), comparing favourably to existing models. Errors on properties related to the potential-energy surface (i.e., U, $U_0$, H, G, and ZPVE) are higher than for other models (Allegro, PaiNN, PaxNet, DimeNet). For the remaining properties, performance is comparable with other state-of-the-art architectures.

Table 3: Mean absolute error (MAE) for the 12 properties of the QM9 dataset (Ramakrishnan et al., 2014). AMP refers to the model proposed in this work. Values for existing models in the literature are taken from: SchNet (Schütt et al., 2018), DimeNet (Klicpera et al., 2020), PaiNN (Schütt et al., 2021), Allegro (Musaelian et al., 2022), SEGNN (Brandstetter et al., 2021), and PaxNet (Zhang et al., 2022).

| Target | Unit | SchNet | DimeNet | PaiNN | Allegro | SEGNN | PaxNet | AMP |
|---|---|---|---|---|---|---|---|---|
| $U_0$ | meV | 14 | 8.02 | 5.85 | **4.7** | 15 | 5.90 | 11.3 |
| U | meV | 19 | 7.89 | 5.83 | **4.4** | 13 | 5.92 | 11.4 |
| H | meV | 14 | 8.11 | 5.98 | **4.4** | 16 | 6.04 | 11.3 |
| G | meV | 14 | 8.98 | 7.35 | **5.7** | 15 | 7.14 | 12.4 |
| ZPVE | meV | 1.7 | 1.29 | 1.28 | - | 1.62 | **1.17** | 4.1 |
| $\epsilon_{HOMO}$ | meV | 41 | 27.8 | 27.6 | - | 24 | **22.8** | 25.7 |
| $\epsilon_{LUMO}$ | meV | 34 | 19.7 | 20.4 | - | 21 | **19.2** | 22.6 |
| $\Delta\epsilon$ | meV | 63 | 34.8 | 45.7 | - | 42 | **31.0** | 44.9 |
| $\mu$ | mD | 33 | 28.6 | 12.0 | - | 23.0 | **10.8** | 11.7 |
| $\alpha$ | $ma_0^3$ | 235 | 46.9 | 45 | - | 60 | **44.7** | 66.8 |
| $\langle R^2 \rangle$ | $ma_0^2$ | 73 | **33.1** | 66 | - | 660 | 93 | 249.2 |
| $c_v$ | $\frac{mcal}{mol \cdot K}$ | 33 | 24.9 | 24 | - | 31 | **23.1** | 31.7 |

## 5 OUTLOOK

In this work, a modified message passing formalism that allows for the efficient incorporation of directional and long-range information was proposed. For future work, an implementation, which makes use of the fast multipole method, as well as the explicit inclusion of many-body interactions, as demonstrated by Klicpera et al. (2020) and Batatia et al. (2022b), and additional weights, as shown by Schütt et al. (2021), could be promising avenues. As a particular use case, we envision the application in (QM)ML/MM MD simulations, which could facilitate the study of large biomolecular systems and enzymatic reactions.

## 6 REPRODUCIBILITY STATEMENT

Code and model parameters used to produce the results in this work are made available at github.com/rinikerlab/AMP. All datasets used in this work are either already publicly available or made available under the same URL.

### ACKNOWLEDGMENTS

The authors thank Lennard Böselt for helpful discussions. The authors gratefully acknowledge NVIDIA for providing a Titan V under the NVIDIA Academic Hardware Grant Program.

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

## A APPENDIX

### A.1 COMPUTATIONAL COST

The modifications proposed in this work will result in additional computational costs. However, it is important to note that the proposed changes will only increase the constant cost of each message but not the overall complexity. The following two paragraphs show empirical running times in a pure QM and in a QM/MM setting. All timings are reported based on a Nvidia Titan V.

Table 4: Relative inference time for a single batch of 100 molecules of the QM9 dataset with a $n$-layer AMP model. The absolute cost is given in parenthesis. The relative cost in percentage is given with respect to the message passing without the proposed modifications as defined in Eq. 1.

| n | AMP(0) | AMP(1) | AMP(2) | AMP(3) |
|---|--------|--------|--------|--------|
| 1 | 138% (5.7 ms) | 146% (6.0 ms) | 164% (6.8 ms) | 207% (8.5 ms) |
| 2 | 151% (8.1 ms) | 164% (8.8 ms) | 200% (10.7 ms) | 267% (14.3 ms) |
| 3 | 162% (10.6 ms) | 178% (11.6 ms) | 221% (14.4 ms) | 307% (20.1 ms) |
| 4 | 167% (12.9 ms) | 187% (14.4 ms) | 236% (18.2 ms) | 333% (25.7 ms) |

### A.1.1 COMPUTATIONAL COST: QM9

Here, the computational cost for various orders and number of layers is reported for a single batch of the QM9 dataset containing 100 samples ($1'833$). This task only includes a single forward pass. Timings were averaged over 100 calls. In addition to absolute costs (in brackets), relative costs with respect to the baseline GNN described in Eq. 1 are reported in Table 4.

### A.1.2 COMPUTATIONAL COST: QM/MM

For QM/MM simulations, which generally include a large number of MM particles and a large number of repeated calls, the cost of the ML potential will significantly contribute to the simulation cost. In general, the computational cost will be dominated by the QM/MM interactions due to the larger number of MM particles. Since the QM/MM interaction is based on a one-time unidirectional pairwise interaction, these costs will be proportional to $N_{QM} \times N_{MM}$, i.e., the number of MM particles times the number of QM particles. The dominating $N_{MM}$ can be reduced further based on the fast multipole method Rokhlin (1985). Here, we report timings for two settings. Setting 1 considers the computational cost without interactions with the MM particles (i.e., the isolated molecule). In setting 2, the computational cost due to the QM/MM interaction is investigated. In both cases, the three layer AMP(3) model used for the systems described in Section 4.3 is used.

In this paragraph, computational costs for the isolated molecule are reported. Batches of size 60 were used. Timings were averaged over all batches using 100 runs and are reported per molecule. A cutoff of 5 Åand three layers were used. Results are reported in Table A.1.2. We find that for the

Table 5: Cost for the prediction of QM energies and QM gradients for the isolated QM systems. The number of particles is shown in the second column. GNN refers to a message passing model based on Eq. 1, i.e., without the proposed modifications.

| System | Size | GNN | AMP(0) | AMP(1) | AMP(2) | AMP(3) |
|--------|------|-----|--------|--------|--------|--------|
| Uracil | 12 | 0.51 ms | 0.65 ms | 0.82 ms | 1.04 ms | 1.52 ms |
| RA | 50 | 0.76 ms | 1.33 ms | 1.53 ms | 2.07 ms | 2.99 ms |
| SAM | 63 | 0.87 ms | 1.67 ms | 1.95 ms | 2.28 ms | 3.15 ms |

two larger systems, the computational cost per edge saturates at roughly 2 $\mu$s per edge as shown in Figure 4. For both molecules, the cost per edge is roughly $50\%$ higher with AMP(3) compared to AMP(2).

Finally, we report the total computational cost to predict energies, QM gradients, and MM-gradients for varying numbers of MM particles in Table A.1.2. Results were averaged over batches of size 10 using 100 runs. Reported are the costs per system.

### A.2 GENERAL IMPLEMENTATION DETAILS

If not noted otherwise, the following architecture and hyperparameters were used. Three message passing layers were used except for the QM9 dataset, where we used four message passing layers. The architecture looked as follows:

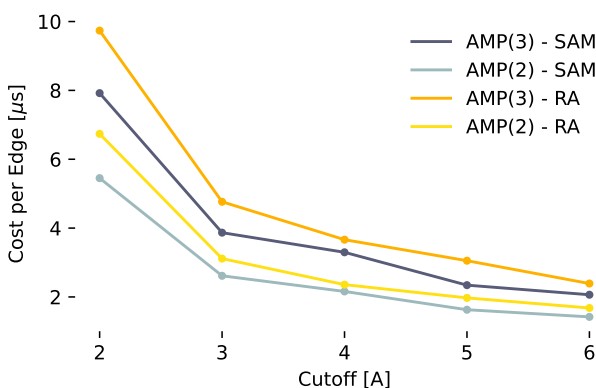

Figure 4: Computational cost per edge for the isolated molecules.

Table 6: Cost for the prediction of QM energies, QM gradients, and MM-gradients for varying numbers of MM particles ($N_{MM}$). Results are reported for the three layer AMP(3) model used in Section 4.3.

| $N_{MM}$ | Uracil | RA | SAM |
|---|---|---|---|
| 10 | 5.6 ms | 20.1 ms | 20.5 ms |
| 100 | 5.7 ms | 22.3 ms | 22.8 ms |
| 1'000 | 8.4 ms | 37.7 ms | 37.4 ms |

1. Generation of edge weights with [Linear(16)], which parametrized $\phi_b$. As input, the onehot features of the two connecting atoms and Bessel function ($n = 20$) expanded distance features were used.

2. Embedding of node features: [Linear(128)]

3. Message layers with [Linear(128), Swish, Linear(128), Swish] for $\phi_h$ and [Linear(128), Swish, Linear(128), Swish, Linear(k+1)] for the prediction of the co-efficients $c_{ij}$, i.e., $\phi_{M(k)}$, $\phi_{M_{QM}(k)}$ and $\phi_{M_{MM}(k)}$. The coefficients for all orders $k$ were predicted with the same module.

4. A readout layer [Linear(128), Swish, Linear(128), Swish, Linear(1)].

Linear($n$) refers to a standard fully connected network layer with bias. No bias was used for the last layer of the output. For each message passing iteration, independent modules were used. Swish was employed as the non-linearity (Ramachandran et al., 2017). Layer weights were initialized based on the method proposed by He et al. (2015). Coefficients $c_{ij}$ were scaled by $10^{-4}$ to support training stability during early epochs.

Node features were initialized as onehot vectors of the element. Norms of the multipoles were concatenated with the hidden feature. Edges were added for all atom pairs within a distance of $5\,\text{Å}$. For interactions between QM and MM particles, an independent long-range cutoff of $50\,\text{Å}$ was used. Radial weights were conditioned on the initial onehot features, i.e., $\phi_b(\mathbf{h}_i^0, \mathbf{h}_j^0, ||\mathbf{r}_{ij}||)$. Distances $||\mathbf{r}_{ij}||$ were encoded with the enveloped Bessel functions proposed by Klicpera et al. (2020) with 20 functions and $p = 5$ for the envelope. The frequencies were not optimized. A ResNet-like update of the hidden feature was used, i.e., $\mathbf{h}_i^{l+1} = \mathbf{h}_i^l + \phi_h(\mathbf{h}_i^l, \mathbf{m}_i)$ (He et al., 2016). Model parameters were optimized using ADAM (Kingma & Ba, 2017) with default parameters ($\beta_1 = 0.9, \beta_2 = 0.999, \epsilon = 10^{-7}$) and an exponentially decaying learning rate ($5 \cdot 10^{-4}, 10^{-5}$). Gradients were clipped by their global norm with a clip norm of 1 (Pascanu et al., 2012). Mean-squared errors were optimized. If the model was jointly trained on energies and gradients, the following loss function was used

$$\mathcal{L} = \frac{(1-\lambda)}{B} \sum_b^B (\hat{V}_b - V_b)^2 + \frac{\lambda}{3BN} \sum_i^{BN} \sum_{\alpha=0}^2 \left( -\frac{\partial \hat{E}_b}{\partial r_{i,\alpha}} - F_{i,\alpha} \right)^2 \tag{12}$$

with the potential energy $V$ and the force component $F_{i,\alpha}$ for each Cartesian dimension $\alpha$. $\lambda = 0.8$ was used to balance the contributions of the energies and gradients to the loss. The loss term for the MM gradients was obtained in the same way but scaled by an additional factor 100 as proposed in (Hofstetter et al., 2022). Models were implemented with TensorFlow (2.6.2 and 2.9.1) (Abadi et al., 2015; Developers, 2021). Single precision `float32` was used.

## A.3  WATER DIMER

Calculations were performed with PSI4 (1.6) (Turney et al., 2012; Parrish et al., 2017; Smith et al., 2020) on a MP2/aug-cc-pVTZ level of theory using density fitted MP2 (Dunlap, 2000; Distasio JR. et al., 2007). The molecules were placed such that the two oxygen atoms were separated by $4\,\text{Å}$. Structures were assigned to training and test sets in an alternating manner. Single layer models were used for all experiments. Models were trained for $10'000$ steps using a batch size of five. 64-dimensional features were used as node features and four edge weights. Since the whole system was simulated with MP2, no partial charge embedding was used. Instead, the pairwise interaction for the model '-DP' was implemented as

$$V_{ESP} = \sum_i \sum_{j > i} \frac{\phi_\rho(\boldsymbol{h}_i) \cdot \phi_\rho(\boldsymbol{h}_j)}{||\boldsymbol{r}_{ij}||} \tag{13}$$

for all long-range pairs, i.e., $||\boldsymbol{r}_{ij}|| \geq r_{\text{cut}}$. Double precision `float64` was used for this system.

### A.3.1  WATER DIMER: LEARNING CURVE

Work on equivariant and directional message passing models has shown that inclusion of directional information can improve data efficiency (Schütt et al., 2021; Batzner et al., 2022; Batatia et al., 2022b). Figure 5 illustrates the convergence of AMP models for the fully connected topology of the water dimer in Figure 2A.

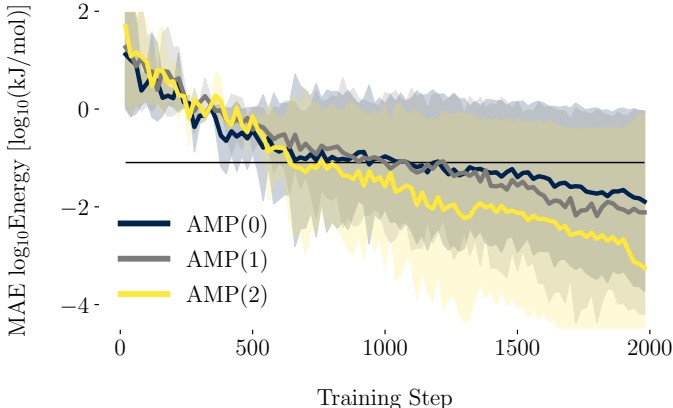

Figure 5: Mean absolute error (MAE) of the energy in $\log_{10}(\text{kJmol}^{-1})$ on the validation set during training for the (fully-connected) dimer topology show in Figure 2A. The bold line indicates the mean over 10 runs while the shaded regions indicate one standard deviation. The black horizontal line indicates a null model (prediction = mean).

In addition, shown in Figure 6, we also explored the training behaviour for the AMP(2) model on the sparsely-connected topology shown in Figure 2B. Interestingly, there appear to be cases where training proceeds in a phase-transition like manner in contrast to the consistent convergence observed for the fully connected topology in Figure 2A.

## A.4  WATER CLUSTERS

The dataset with the water clusters was created based on the geometries presented in (Rakshit et al., 2019) For each set of clusters ranging from 6 to 20 water molecules, up to $2'000$ clusters were randomly sampled. For each cluster, one water molecule was randomly designated as the QM zone with

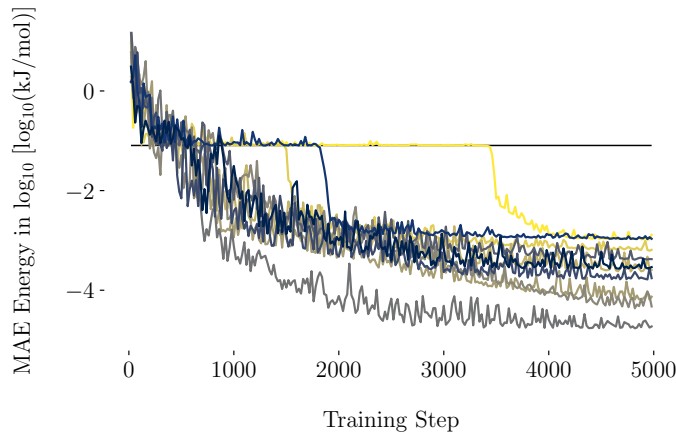

Figure 6: MAE of the energy in $\log_{10}(\text{kJmol}^{-1})$ on the validation set during training for the (sparsely connected) dimer topology show in Figure 2B. Each line represents the training trajectory of one of 10 randomly initialized models. The black horizontal line indicates a null model (prediction = mean).

the remaining molecules forming the MM zone. The long-range cutoff was set to $30\,\text{Å}$, including all QM–MM interactions. All accessible clusters were used, i.e., all known clusters within a 5 kcal/mol energy window above the putative minimum. All clusters of size $n = 10$ were used for training while the other clusters were used for testing. For the PaiNN, SchNet, and GNN models, the energy was predicted for the QM particles. Furthermore, in addition to the existing atom types (oxygen-QM and hydrogen-QM), two new types were introduced: oxygen-MM and hydrogen-MM. The implementation provided in the SchNetPack (Schütt et al., 2019) was used for PaiNN and SchNet (Schütt et al., 2021; Schütt et al., 2018). For PaiNN, the size of the scalar representation (160) was increased to provide a comparable number of trainable parameters ($1M$). For SchNet, the size of layers and the number of filters was set to 256. The same cutoff (5 Å), readout and radial basis functions were used as in the AMP model. Numerical results for all models are reported in Table A.4.1. Partial charges of the MM zone were assigned based on MBIS monopoles (Verstraelen et al., 2016), i.e., $-0.914197e$ for oxygen and $0.457098e$ for hydrogen. Calculations were performed on a MP2/aug-cc-pVTZ level of theory (Møller & Plesset, 1934; Dunning, 1989; Kendall et al., 1992) using density fitted MP2 (Dunlap, 2000; Distasio JR. et al., 2007) as implemented in PSI4 (1.6) (Turney et al., 2012; Parrish et al., 2017; Smith et al., 2020). Gradients and energies of the isolated QM zone were subtracted. The models were trained over $50'000$ steps, presenting each time a batch of 16 randomly drawn samples.

### A.4.1 WATER CLUSTERS: RESULTS

Table 7: Mean absolute error (MAE) of the energies in kJ mol$^{-1}$ presented in Figure 3.

| Cluster Size | 6 | 7 | 8 | 9 | 10 | 11 | 12 | 13 | 14 | 15 | 16 | 17 | 18 | 19 | 20 |
|---|---|---|---|---|---|---|---|---|---|---|---|---|---|---|---|
| AMP(0) | 7.8 | 6.25 | 3.97 | 5.4 | 5.11 | 5.76 | 4.4 | 5.14 | 5.06 | 5.13 | 4.67 | 4.71 | 5.05 | 5.02 | 5.11 |
| AMP(1) | 2.66 | 2.22 | 1.44 | 0.77 | 0.25 | 0.53 | 0.86 | 0.92 | 1.02 | 1.36 | 2.03 | 3.01 | 4.59 | 6.2 | 8.1 |
| AMP(2) | 0.3 | 0.33 | 0.17 | 0.18 | 0.16 | 0.26 | 0.23 | 0.26 | 0.28 | 0.34 | 0.41 | 0.42 | 0.47 | 0.56 | 0.55 |
| AMP(3) | 0.4 | 0.39 | 0.17 | 0.23 | 0.16 | 0.33 | 0.33 | 0.39 | 0.46 | 0.52 | 0.62 | 0.8 | 0.82 | 0.99 | 0.86 |
| PaiNN | 1.14 | 1.03 | 0.44 | 0.51 | 0.08 | 0.75 | 0.65 | 0.8 | 0.89 | 1.01 | 1.2 | 2.08 | 1.91 | 2.19 | 1.8 |
| SchNet | 1.79 | 1.42 | 0.6 | 0.75 | 0.19 | 1.05 | 1.02 | 1.28 | 1.37 | 1.5 | 1.75 | 3.61 | 3.61 | 4.1 | 4.11 |
| GNN | 6.22 | 5.09 | 1.47 | 1.48 | 0.07 | 1.68 | 2.18 | 3.01 | 3.63 | 4.34 | 4.8 | 7.54 | 6.74 | 6.94 | 6.94 |
| GNN-P | 1.16 | 0.88 | 0.37 | 0.39 | 0.09 | 0.57 | 0.54 | 0.8 | 1.0 | 1.2 | 1.34 | 2.32 | 2.62 | 3.43 | 3.6 |

### A.5 QM/MM SYSTEMS

The datasets were taken from Böselt et al. (2021) and randomly split into training/validation/test sets of $7'000/2'000/1'000$ samples, respectively. Polarization due to the MM particles described in Eq. 10 was only incorporated during the last message passing step. In addition, only eight distance

Table 8: Mean absolute errors (MAE) of the gradients in kJ mol$^{-1}$ Å$^{-1}$ presented in Figure 3.

| Cluster Size | 6 | 7 | 8 | 9 | 10 | 11 | 12 | 13 | 14 | 15 | 16 | 17 | 18 | 19 | 20 |
|---|---|---|---|---|---|---|---|---|---|---|---|---|---|---|---|
| AMP(0) | 8.04 | 7.92 | 8.48 | 8.27 | 8.4 | 8.14 | 8.35 | 8.2 | 8.2 | 8.23 | 8.23 | 8.28 | 8.29 | 8.48 | 8.52 |
| AMP(1) | 1.65 | 1.37 | 0.76 | 0.61 | 0.18 | 0.81 | 1.18 | 1.59 | 2.09 | 2.72 | 3.68 | 5.48 | 6.73 | 8.08 | 9.11 |
| AMP(2) | 0.58 | 0.52 | 0.26 | 0.26 | 0.12 | 0.36 | 0.32 | 0.43 | 0.5 | 0.6 | 0.73 | 0.91 | 1.06 | 1.25 | 1.3 |
| AMP(3) | 0.67 | 0.59 | 0.25 | 0.3 | 0.1 | 0.48 | 0.44 | 0.64 | 0.8 | 1.01 | 1.23 | 1.32 | 1.61 | 1.98 | 2.19 |
| PaiNN | 1.0 | 1.06 | 0.5 | 0.61 | 0.05 | 0.85 | 0.68 | 0.77 | 0.77 | 0.79 | 0.88 | 1.56 | 1.44 | 1.52 | 1.35 |
| SchNet | 2.26 | 1.84 | 0.7 | 0.97 | 0.12 | 1.34 | 0.96 | 1.16 | 1.18 | 1.18 | 1.3 | 2.4 | 2.36 | 2.57 | 2.52 |
| GNN | 3.42 | 2.99 | 1.01 | 1.43 | 0.07 | 2.02 | 1.55 | 1.89 | 1.98 | 2.05 | 2.24 | 3.94 | 3.71 | 3.8 | 4.0 |
| GNN-P | 1.76 | 1.49 | 0.68 | 0.73 | 0.05 | 0.97 | 0.95 | 1.29 | 1.61 | 1.95 | 2.22 | 2.74 | 3.21 | 3.78 | 3.86 |

weights were used for interactions between QM and MM particles while the default of 16 was used for edges within the graph, i.e., between QM particles. The models were trained over $400'000$ steps using batches of size 5 for uracil and batches of size 1 for retinoic acid and SAM/cysteine. For the comparison between the long-range cutoffs, only $200'000$ training steps were used.

## A.6 QM9 DATASET

The dataset was taken from Ramakrishnan et al. (2014). An independent 4-layer AMP(3) model was trained for each property over $1'250$ epochs with an exponentially decaying learning rate ($5 \cdot 10^{-4}$, $5 \cdot 10^{-6}$). A 4-layer AMP(3) results in $1'153'793$ trainable parameters. Properties in units of $Eh$ were converted and trained in units of kcal mol$^{-1}$. Each property was predicted as the sum of atomic contributions except for the dipole moment $\mu$. For all properties, except for the dipole moment $\mu$, the mean of the training set was subtracted. In addition, the energy of the isolated atoms was subtracted for $U_0$, U, H, and G. $3'054$ molecules for which the geometry consistency checks failed were removed from the dataset Ramakrishnan et al. (2014). The remaining data points were randomly split into a training/validation/test sets ($110'000$, $10'000$, $10'831$) using a batch size of 100. As proposed by Veit et al. (2020), the molecular dipole moment $\boldsymbol{\mu}$ was modeled as the sum of the contribution from atomic monopoles $q_i$ and atomic dipoles $\boldsymbol{\mu}_i$

$$\boldsymbol{\mu} = \sum_i \phi_q(\boldsymbol{h}_i^n)\boldsymbol{r}_i + \boldsymbol{\mu}_i \tag{14}$$

with each atomic dipole $\boldsymbol{\mu}_i$ being predicted according to Eq. 3. The mean excess charge was subtracted from each $q_i$ to enforce charge conservation.

