# OpenReview forum: "Anisotropic Message Passing: Graph Neural Networks with Directional and Long-Range Interactions"
_ICLR.cc/2023/Conference — ICLR 2023 poster_

### Official Review · Reviewer_yByC · 2022-10-21

**Confidence:** 3
**Clarity, Quality, Novelty And Reproducibility:** The quality of the whole paper is goo…
**Correctness:** 4
**Technical Novelty And Significance:** 3
**Empirical Novelty And Significance:** 3
**Recommendation:** 6

**Strength And Weaknesses:**

Strength:
1. The introduction of QUANTUM MECHANICS/MOLECULAR MECHANICS sees some novel.
2. Experimental results on WATER DIMERS show the effectiveness of the proposed model.

Weakness:

1. The description of the developed methodology is unclear. It is hard to follow the work. The notation and equation formulation are complicated. Together with the corresponding explanation, they should be largely improved.
2. The long range information is the main advantage of the proposed method, however, there are no corresponding experimental discuss or theoretical analyses in the manuscript.
3. The computation complexity analysis is a big concern. The proposed method uses more parameters and computation steps during message passing, this would bring lots of computation burden. However, from the real dataset experimental results in table 4, the performance promotion is slight.
4 The novelty isn't particularly high. Most techniques used were proposed and demonstrated before.
5. How about experimental results on the PDBbind dataset.

**Summary Of The Paper:**

This paper propeses a  modified message passing GNN that allows for the efficient incorporation of directional and long-range information. Based on the anisotropic feature, a new message passing functions are designed.

**Summary Of The Review:**

The view of the idea for GNN is novel. The theoretical modeling seems meaningful for some datasets. However, this also brings the underlying computation burdens, especially considering limited performance promotion.

---

> ### Author Response · Authors · 2022-11-17
> **Response to Reviewer 4 (yByC)**
>
> > W1: The description of the developed methodology is unclear. It is hard to follow the work. The notation and equation formulation are complicated. Together with the corresponding explanation, they should be largely improved.
>
> A1: We are sorry to hear that the description of the work was difficult to follow. As the other reviewers considered the manuscript overall well-written, we would need more concrete pointers to where the formalism was not clear enough in order to improve them. Could you please be more specific where you would like to have more clarity/explanations?
>
> > W2: The long range information is the main advantage of the proposed method, however, there are no corresponding experimental discuss or theoretical analyses in the manuscript.
>
> A2: In “Results and Discussion”, the three subsections 4.1 (with the -DP model), 4.2 (all interactions with the MM particles are long-range in nature since the MM particles are not part of the graph), and 4.3 (comparison with 30A cutoff vs 50A cutoff) study long-range interactions. The fact that the model can learn energies and gradients for the water clusters and the QM/MM systems is in our opinion a clear evidence that the model can learn long-range interactions since the MM particles are not part of the graph and are at times more than 40 Angstroms away from the QM particles. From a theoretical viewpoint, Section 3.6 discusses how we incorporate long-range interactions (i.e., through a polarization term and a pairwise electrostatic interaction).
>
> > W3: The computation complexity analysis is a big concern. The proposed method uses more parameters and computation steps during message passing, this would bring lots of computation burden. However, from the real dataset experimental results in table 4, the performance promotion is slight. 4 The novelty isn't particularly high. Most techniques used were proposed and demonstrated before.
>
> A3: This question was also raised by the other reviewers. We have added more results concerning the run time in the Appendix. The time complexity is not different compared to standard message passing: the modifications only increase the constant cost of each message. We hope that this new section addresses the concerns. Generally, computational costs are small compared to the other models discussed in Section 4.4 for which computational costs were reported. PaiNN is the only model for which computational cost/accuracy is comparable.
>
> > 4. How about experimental results on the PDBbind dataset.
>
> A4: We thank you the reviewer for this suggestion. This would definitely be an interesting case, particularly due to the long-range interactions present in such systems, and we look into this in future work. However, it is out of scope for the current study.

---

### Official Review · Reviewer_2uEo · 2022-10-24

**Confidence:** 3
**Correctness:** 3
**Technical Novelty And Significance:** 3
**Empirical Novelty And Significance:** 3
**Recommendation:** 6

**Clarity, Quality, Novelty And Reproducibility:**

Clarity:Good: The paper is well organized, but the presentation has minor details that could be improved.

Quality:Good: The paper appears to be technically sound. The proofs, if applicable, appear to be correct, but I have not carefully checked the details. The experimental evaluation, if applicable, is adequate, and the results convincingly support the main claims.

Novelty:Fair: The paper contributes some new ideas or represents incremental advances.

Reproducibility: Good: key resources (e.g., proofs, code, data) are available.

**Strength And Weaknesses:**

1. Strength:

S1. The paper is technically sound.

S2. Extensive experiments are designed to verify the performance of the proposed method.

S3. The paper is well-written and easy to follow.

2. Weaknesses:

W1. There are many other GNN models. I other GNN models? Can you be more clear? I am not sure which GNN model you used.

W2. I noticed that the performance for "shuffled" data in Table 2 is relatively bad. Can you explain it? Is there any way to improve it? Since we always use shuffled data for training and test to evaluate deep learning models.

W3. Can you offer the time complexity analysis or add more experiments on the running time?

**Summary Of The Paper:**

This paper focused on the message-passing schemes of a GNN model. It integrates the anisotropic state based on Cartesian multiples to the message-passing models to compensate for the hidden features. Such operation enhances the interactions, i.e., (1) anisotropic long-range, (2) the surrounding fields and particles not part of the graph.

**Summary Of The Review:**

In general, it is a  technically sound paper, which proposes a novel message-passing scheme in the Quantum Chemistry fields.

---

> ### Author Response · Authors · 2022-11-17
> **Response to Reviewer 3 (2uEo)**
>
> > W1. There are many other GNN models. I other GNN models? Can you be more clear? I am not sure which GNN model you used.
>
> A1: We assume that you refer to GNN and GNN-P in Table 1. We used the label 'GNN' for message passing as described in equation 1, i.e. the exact same model without the proposed anisotropic feature/state. However, since we replaced the baseline GNN with PaiNN, this part was removed, which we hope makes this more clear.
>
> > W2. I noticed that the performance for "shuffled" data in Table 2 is relatively bad. Can you explain it? Is there any way to improve it? Since we always use shuffled data for training and test to evaluate deep learning models.
>
> A2: This is indeed an important observation (we assume that you meant 'unshuffled' seems bad). There are probably a few contributing factors:
> The dataset is challenging, due to (a) a large QM system (including intra- and intermolecular interactions in the case of SAM), (b) a larger number of MM particles, and (c) the fact that the dataset consists of a relatively short (3.5ps + 1.5ps) continuously sampled trajectories. The latter point means that there is a large degree of redundancy/correlation within the training set.
> Results for most models in the literature use relatively small/simple systems, which might give the wrong impression of the current state of ML potentials. Although the strict temporal script used in 'unshuffled' is rarely used in the literature (e.g. the often reported MD17 is typically used in a 'shuffled' setting) but represents a possible use case for larger systems (where it is not as easy to generate diverse training data points). Therefore, we think that it is worth including results on challenging systems, which we hope will also inform further research directions.
>
> There are different strategies to improve this (e.g. improve the sampling of the training data points, simplify the learning task, or physical priors), but they are very much part of ongoing research.
>
> > W3. Can you offer the time complexity analysis or add more experiments on the running time?
>
> A3: This is an important point, which was also raised by the other reviewers. We have added more experiments concerning the run time in the Appendix. Regarding the time complexity: For the QM setting, the AMP model will inherit the time complexity of general message passing models (which will be somewhere between O(N) and O(N**2) depending on the cutoff (where N is the number of particles)). In the QM/MM setting, costs will be dominated by the interaction with the MM charges, which will be O(N_QM * N_MM). An implementation of the fast multipole method for this interaction could decrease N_MM.

---

### Official Review · Reviewer_YdbF · 2022-10-24

**Confidence:** 4
**Correctness:** 4
**Technical Novelty And Significance:** 4
**Empirical Novelty And Significance:** 4
**Recommendation:** 8

**Clarity, Quality, Novelty And Reproducibility:**

The paper implicitly assumes several computational chemistry ideas (which were not easy to follow for me, an outsider), but well written in total. The paper covers relevant literature about ML potentials and ML for the simulation of QM/MM systems. They also provide the code base, and the idea is clean and simple. Thus the paper's results and conclusions would be reproducible. I don't know much about the related literature on ML potentials / ML+simulation / Physics-/Chemistry-informed ML published in non-ML venues (such as chemistry or chemoinfo journals), but as far as I know, the idea of this paper is novel.

I think I understood the point of the last sentence on Page 3 "Albeit possible, they do not necessarily carry physical meaning." But further investigations and clarifications on this point would be a promising research topic. It was very interesting to me that the main idea of this paper comes from standard techniques (multipole expansions) in computational chemistry, but it is just prepared through learning parameters, and indeed, "they do not necessarily carry physical meaning" (Nevertheless, it worked apparently)

Three minor comments/questions on details:

- p.3, "r_^0_{ij} = 1" below eq(2) should be "R_^0_{ij} = 1"...?

- p.4 eq (8), a_{ij} in the first equation should be "u_{ij}"...?

- AMP(k) can be worse than AMP(k+1) because increasing k means increasing the number of model parameters (and might make the learning harder)...?

**Strength And Weaknesses:**

[Strength]

- The proposed scheme of Anisotropic Message Passing (AMP) is an interesting and natural scheme to consider any directional effects from an external field or any long-range effects from a set of particles unreachable in the graph used for message passing. When we use any 3D geometric GNN, message passing is often done within cutoff ranges rather than complete graph having edges between all pairs of nodes for efficient computation. But this ignores long-range interactions between atoms or particles. Not only this is suitable to considering the interactions to an external field such as QM/MM or multi-molecules cases, but also suitable for a single molecule case if we use distance-based edge cutoff (as demonstrated in QM9 cases of 4.4).

- The idea to introduce anisotropy is based on multipole expansions with respect to introduced "internal anisotropic states", which sounds convincing and neat. It would be well motivated when we consider such directional/angular effects in computational chemistry. The use of series expansion sounds well suited because, similarly to Taylor series, the first term has a good approximation of the original function, and higher-order terms are correction terms to represent deviations of the approximation due to the k-order directional effects (we can control the approximation precision by setting the order k). Also, the use of such anisotropic states in message passing corresponds to multipole-multipole interactions that are also well discussed in the computational chemistry literature as cited.

- The paper comes with three nice case-study demonstrations: (i) 2 of water molecules (water dimers), (ii) QM/MM calculations of 6-20 water molecules (water cluster), (iii) five molecules of varying size solvated in water (a realistic QM/MM system example from Böselt et al. 2021), (iv) QM9 dataset with edge cutoff of 5 angstrom.


[Weaknesses]

- As we can see in Figure 2 (water dimer cases), if we have a complete graph (edges are formed between all nodes), then we don't need to use this AMP and just use standard message passing. So I felt that one of big values of this paper would be practical efficiency. Given that we often apply distance cutoff for geometric message passing, this point is also appealing, but the paper can have any discussions and concrete results on how inefficient if we consider a complete graph for message passing.

- "directional interactions" in the title was easy to get, but "long-range interactions" in the title were unclear to me at first glance. When we use a 3D geometric GNN such as SchNet and DimeNet (unlike 2D non-geometric GNNs), we can apply message passing to all pairs of nodes in theory. We don't necessarily need to apply distance cutoff to edges if the practical efficiency is OK (in the case of small molecules?). It would be nice to have descriptions on this to make clear this point.



**Summary Of The Paper:**

This paper considers a 3D geometric GNN, and develops a novel message passing scheme considering the directional effects from an external field or from a set of particles unreachable in a graph. If we place a molecule in an electric field, we need to take it into account when calculating the potential energies of the molecule. The developed scheme is a perfect fit to the cases, for example, (A) a geometric graph in R^3 with edge connections by distance cutoff, where long-range interactions beyond the cutoff are ignored during message passing, and (B) QM/MM situations where we consider a set of particles in quantum mechanics (QM), but other particles treated in molecular mechanics (MM). The idea for this is to prepare a state tensor (anisotropic state) M_i^k for each node i, and used in message passing. M_i^k is defined by linear combinations of basis, i.e., the tensor product of order k of the unit vector from node_i to a neighbor j, where coefficients are learned by neural nets. This corresponds to multipole expansions of an angle-dependent function on R^3, which is often used in computational chemistry (monopoles, bipoles, dipoles, ...). The paper also provides case studies of water dimer cases (interactions between 2 H2O molecules) and water cluster (a set of many water molecules) as well as QM9 benchmarking with geometric graphs with an edge cutoff of 5 angstrom.

**Summary Of The Review:**

Basically, I liked the paper. In order to consider the effects from an external field (directional interactions) or those from unreachable nodes (long-range interactions), this paper develops a novel message passing scheme called Anisotropic message passing. The directional or long-range contributions are formulated by a multipole series expansion with learnable coefficients. The use of series expansion is also a good fit for me. Then the entire long-range or directional effects are modeled via message passing with multipole-multipole interactions as the proposed Anisotropic Message Passing (AMP) scheme of order k. Also, the empirical evaluations with four nice use cases are very convincing.

---

> ### Author Response · Authors · 2022-11-17
> **Response to Reviewer 2 (YdbF)**
>
> > Q1: As we can see in Figure 2 (water dimer cases), if we have a complete graph (edges are formed between all nodes), then we don't need to use this AMP and just use standard message passing. So I felt that one of big values of this paper would be practical efficiency. Given that we often apply distance cutoff for geometric message passing, this point is also appealing, but the paper can have any discussions and concrete results on how inefficient if we consider a complete graph for message passing.
>
> We think there are two points worth discussing here:
> 1) Empirically, GNNs seem to perform worse when the distance cutoff is increased beyond a certain point, with most models using a cutoff around 5 Angstrom. This observation in the setting of computational chemistry is very likely related to the reference ‘oversquashing', which was reported in recent ML research [1, 2]. Using a shorter cutoff could thus also simplify the learning task.
> 2) Regarding the computational cost: We have added a section in the Appendix where we report the computational cost for three settings (QM9, QM systems, and QM/MM systems). We hope that this clarifies this point. We find that for sufficiently large graphs, the computational cost is O(E), where E is the number of edges. This means that we should expect roughly 50% less computational cost (15 vs 7 edges).
>
> [1] Uri Alon and Eran Yahav. On the bottleneck of graph neural networks and its practical implications. arXiv, pp. arXiv:2006.05205, 2020.
>
> [2] Vijay Prakash Dwivedi, Ladislav Rampasek, Mikhail Galkin, Ali Parviz, Guy Wolf, Anh Tuan Luu, and Dominique Beaini. Long range graph benchmark. arXiv, pp. arXiv:2206.08164, 2022
>
> > Q2: "directional interactions" in the title was easy to get, but "long-range interactions" in the title were unclear to me at first glance. When we use a 3D geometric GNN such as SchNet and DimeNet (unlike 2D non-geometric GNNs), we can apply message passing to all pairs of nodes in theory. We don't necessarily need to apply distance cutoff to edges if the practical efficiency is OK (in the case of small molecules?). It would be nice to have descriptions on this to make clear this point.
>
> Regarding the long-range interactions: This is a really important observation and we have added small modifications in the text to make this clearer. In physics and chemistry, “long-range” generally refers to interactions, which scale as 1/R, and which, as a result, do not convergence in 'real-space' (compared to reciprocal space), or interactions, which convergence very slowly in ‘real-space'. In the ML setting, this notion seems to be less precise: For GNN, “long-range” seems to refer to pair of nodes, which cannot exchange messages at all or only within multiple iterations. It is correct that we do not need to apply a cutoff in principle, however, as mentioned in point 1, there seems to be strong empirical evidence that a cutoff is advantageous, beyond simply decreasing computational cost.
>
> Following the suggestion, we had another look at the learning curves for these dimer test-systems, which are now shown in the Appendix (3.1), and made an interesting observation. We have found that for the fully connected topology, higher-order models converge faster to a lower error. Interestingly, the convergence differs qualitatively for the sparsely connected topology (Figure 2B), while the convergence is more consistent for the fully connected topology. However, for the sparse topology, convergence seems to proceed in a ‘phase-transition-like' manner. We think that this observation can be explained with the redundancy of information in the former setting, while the latter setting rests on a single piece of information (i.e. the quadrupole-quadrupole component). Initialisation probably also plays an important role in this case. We think that this shows nicely that different topologies not only affect computational cost but also the error landscape.
>
> > Three minor comments/questions on details:
> >- p.3, "r_^0_{ij} = 1" below eq(2) should be "R_^0_{ij} = 1"...?
> >- p.4 eq (8), a_{ij} in the first equation should be "u_{ij}"...?
> >- AMP(k) can be worse than AMP(k+1) because increasing k means increasing the number of model parameters (and might make the learning harder)...?
>
> We thank the reviewer for the comments and suggestions:
> a) This was indeed wrong and we have corrected it.
> b) This is correct but maybe needs further clarification. We suggest here that we can also use the anisotropic features to refine the anisotropic state itself; for 'bootstrapping' we use simply the distance feature during the first iteration. We have adapted the text to make this point more clear.
> c) This is a sound explanation. Its also important to note that we did not include any example where AMP(3) is well motivated (e.g. transition metals).

---

> ### Comment · Reviewer_YdbF · 2022-11-22
> **I have acknowledged that I read the response and updates**
>
> Thanks for the clarification and sorry for this late post. It seemed that I got the discussion period wrong because the initial email to reviewers says Discussion period: Nov 4 2022 - Dec 12 2022 and I missed the schedule update notice. PC told me that authors can still participate in the discussion till Dec 12, and I'll leave my comment here.
>
> The responses made my questions clear, and the corresponding updates were informative to me. In particular, as in the "General Comment / Revision" post, adding the report in Appendix about the computational cost for various settings and system sizes (QM9, QM/MM) and comparisons to PaiNN (main-text) and SchNet (appendix)  is really nice to see.
>
> The update of explicitly mentioning the meaning of "long-range" also clarifies the paper's intention. I felt that wording "long-range interactions" in ML community is associated with attention, in particular, Transformers, as we see in "LambdaNetworks: Modeling Long-Range Interactions Without Attention (ICLR 2021)" or "Focal Attention for Long-Range Interactions in Vision Transformers" (NeurIPS 2021).

---

### Official Review · Reviewer_cgSu · 2022-10-25

**Confidence:** 4
**Correctness:** 2
**Technical Novelty And Significance:** 2
**Empirical Novelty And Significance:** 2
**Recommendation:** 6

**Clarity, Quality, Novelty And Reproducibility:**

The clarity and quality of the proposed method are good. The novelty is fair. The authors provide code and model parameters to reproduce the results.

**Strength And Weaknesses:**

Pros:
1. The paper is well-organized.
2. The paper proposes a practical way to address long-range and directional interactions for QM/MM systems.

Cons:
1. The novelty of the proposed model is limited. There are existing works [1-4] using GNN to deal with QM/MM systems or force fields. There is also work [5] addressing multiples using GNN.
2. In the experiments, the proposed model is only compared to limited or simple baselines without including enough related and state-of-the-art models.
3. For the experiment about water dimers, the seems to be no difference between the curves when k=1 and k=2 in Figure 2A. However, the authors mention that "the learning task seems to be more challenging for the AMP(1) compared to the AMP(2) model, indicated by marginally larger errors", which is misleading. Moreover, the authors claim that Dimenet and PaiNN "seem to fail to describe this quadrupole-quadrupole interaction" without having convincing experimental results. Lastly, the authors use the wrong name ("Dime") when referring to DimeNet.
4. For the experiment about QM/MM systems, the comparison to the previous models is unclear. For example, Table 2 and 3 only show the results of the proposed model without having the results of other models.
5. For the experiment on QM9 dataset, I'm feeling that this experiment is not very suitable to be shown here, since QM9 is mainly for the prediction of scalar molecular properties. I would suggest using MD17, which is a commonly used benchmark dataset that includes MD trajectories of small organic molecules. The goal is to predict energies and forces, which is more suitable for the proposed model.
6. Although the authors claim that their proposed modifications require little additional computational cost, there are no empirical results addressing that.

[1] Qiao, Zhuoran, et al. "Informing geometric deep learning with electronic interactions to accelerate quantum chemistry." Proceedings of the National Academy of Sciences 119.31 (2022): e2205221119.

[2] Haghighatlari, Mojtaba, et al. "Newtonnet: A newtonian message passing network for deep learning of interatomic potentials and forces." Digital Discovery (2022).

[3] Unke, Oliver T., et al. "SpookyNet: Learning force fields with electronic degrees of freedom and nonlocal effects." Nature communications 12.1 (2021): 1-14.

[4] Batzner, Simon, et al. "E (3)-equivariant graph neural networks for data-efficient and accurate interatomic potentials." Nature communications 13.1 (2022): 1-11.

[5] Thürlemann, Moritz, Lennard Böselt, and Sereina Riniker. "Learning atomic multipoles: Prediction of the electrostatic potential with equivariant graph neural networks." Journal of Chemical Theory and Computation 18.3 (2022): 1701-1710.

**Summary Of The Paper:**

In this paper, a message passing GNN is proposed by incorporating an anisotropic state based on Cartesian multiples to address long-range and directional interactions in chemical systems. The proposed model is evaluated on two model systems and two existing datasets for quantum mechanics/molecular mechanics (QM/MM) systems and small organic molecules.

**Summary Of The Review:**

Given that there are several existing papers addressing a similar question, the authors should cover and discuss these related works. The experiments should be better designed with more compared baselines and more suitable datasets.

----------------------------------------------------------------------
Based on the authors' response:
The authors have addressed my concerns about the related works, comparisons to other models, computational cost, and dataset. The updated version has been improved to better demonstrate the proposed method.

---

> ### Author Response · Authors · 2022-11-17
> **Response to Reviewer 1 (cgSu) (1-3)**
>
> > Q1: The novelty of the proposed model is limited. There are existing works [1-4] using GNN to deal with QM/MM systems or force fields. There is also work [5] addressing multiples using GNN.
>
> A1: We feel that there might be a misunderstanding. We are aware that a large number of ML potentials/force-fields were proposed in recent years, however, none of the references [1-4] report results for a QM/MM system and only [1] mentions QM/MM in one sentence. In addition, of the references [1-4] only [4] reports results for condensed-phase systems. QM/MM with electrostatic embedding poses specific requirements to a ML model that we try to explore and address in this work. At the same time, we also aim to highlight QM/MM as a powerful formalism ideally suited for ML, with the hope to inspire further research in this direction. Regarding reference [5]: The present work builds on this reference. The difference is that ref [5] was specifically developed to predict atomic multipoles trained on given reference multipoles. In the present work, multipoles serve as a tool to introduce directional information and do not necessarily carry a physical meaning. In addition, the anisotropic features, which were not present in [5], can be used to improve the accuracy of the model proposed in [5].
>
> > Q2: In the experiments, the proposed model is only compared to limited or simple baselines without including enough related and state-of-the-art models.
>
> A2: We thank the reviewer for raising this point. For the test systems in Section 4.1 and 4.2, we aim to motivate the proposed changes qualitatively. Following your suggestion, we have now included the state-of-the-art model PaiNN [6] as the baseline for the water clusters.
>
> > Q3: For the experiment about water dimers, the seems to be no difference between the curves when k=1 and k=2 in Figure 2A. However, the authors mention that "the learning task seems to be more challenging for the AMP(1) compared to the AMP(2) model, indicated by marginally larger errors", which is misleading. Moreover, the authors claim that Dimenet and PaiNN "seem to fail to describe this quadrupole-quadrupole interaction" without having convincing experimental results. Lastly, the authors use the wrong name ("Dime") when referring to DimeNet.
>
> A3: (a) The curve shown in Figure 2A seems indeed to suggest that both models are equivalent, while the MAE is slightly lower for AMP(2) compared to AMP(1). We have now rephrased the corresponding sentence to make it clearer. We also note that this interpretation is consistent with the findings presented previously in the literature, e.g. in Figure 4 of the PaiNN publication [6] or in Figure 7 of the Nequip publication [7]. Furthermore, we have added learning curves (Appendix 3.1) for this setting, which we think support our conclusions.  (b) Our statements for the quadrupole interaction (Figure 2b) are based on analytical/theoretical arguments. DimeNet cannot describe this interaction because it uses angles between edges to introduce directional information; however all angles between edges are invariant under this specific rotation. Similarly, the equivariant feature vector of PaiNN, which can conceptually be understood as a 'high-dimensional dipole' (i.e. k=1) cannot describe this interaction as it is also invariant under the considered rotation. We have tried to train PaiNN on this system (with a cutoff of 4.25A), but without success. We also had a look at the learning curves with the PaiNN model, which we could not consistently converge. We have decided against showing these results because we do not know the PaiNN architecture as well as our own model, which could result in an unfair comparison.  (c) We thank the reviewer for spotting the wrong naming, we have corrected it.

---

> ### Author Response · Authors · 2022-11-17
> **Response to Reviewer 1 (cgSu) (4-6)**
>
> >Q4: For the experiment about QM/MM systems, the comparison to the previous models is unclear. For example, Table 2 and 3 only show the results of the proposed model without having the results of other models.
>
> A4: We thank the reviewer for raising this point. Indeed, we only show the comparison between different hyperparameters/training regimes for the AMP model. We did not make a direct comparison with the previous work because they did not report the results for the RA and SAM systems without delta learning. However, they reported the results for the Uracil system without delta learning, which we have included now.
>
> > Q5: For the experiment on QM9 dataset, I'm feeling that this experiment is not very suitable to be shown here, since QM9 is mainly for the prediction of scalar molecular properties. I would suggest using MD17, which is a commonly used benchmark dataset that includes MD trajectories of small organic molecules. The goal is to predict energies and forces, which is more suitable for the proposed model.
>
> A5: We chose QM9 as a comparative benchmark mainly because it has been extensively used as a benchmark in previous work, thus offering a general impression of the 'power' of the representation compared to existing models. Energies and gradients are investigated with all other experiments in our manuscript. We considered it worthwhile to also investigate properties, since dipoles (mu) and polarizabilities (alpha) play an important role in practical applications of QM/MM simulations (e.g. for the calculation of vibrational spectra). We would also like to point out that we predicted the molecular dipole as a vectorial property. In our opinion, MD17 would not add a new perspective, since performances between QM9 and MD17 are strongly correlated for existing systems. It is also important to note that MD17 is similarly limited as QM9 [see e.g. 8].
>
> > Q6: Although the authors claim that their proposed modifications require little additional computational cost, there are no empirical results addressing that.
>
> A6: We thank the reviewer for this suggestion. We have added additional results regarding the computational costs for QM9, isolated QM systems, and QM/MM systems in the Appendix. The statement referred to the computational cost we had reported for a batch of the QM9 system (which has been reported by previous publications) and the general benefits the model offers since it does not require inclusion of MM particles in the graph.
>
> [6] Kristof T. Schütt, Oliver T. Unke, and Michael Gastegger. Equivariant message passing for the prediction of tensorial properties and molecular spectra. In International Conference on Machine Learning, pp. 9377–9388. PMLR, 2021
>
> [7] Simon Batzner, Albert Musaelian, Lixin Sun, Mario Geiger, Jonathan P. Mailoa, Mordechai Kornbluth, Nicola Molinari, Tess E. Smidt, and Boris Kozinsky. E(3)-equivariant graph neural networks for data-efficient and accurate interatomic potentials. Nat. Commun., 13:2453, 2022
>
> [8] J. Bowman et al, The MD17 datasets from the perspective of datasets for gas-phase ‘small’ molecule potentials, J. Chem. Phys. (2022)

---

> ### Comment · Reviewer_cgSu · 2022-11-26
> **Response to Authors**
>
> I would like to thank the authors for addressing my points raised. The updated manuscript is better than the previous one by reporting the computational cost and adding more comparisons to the baselines. Based on these, I would like to raise my score. However, I suggest the authors add a theoretical analysis of the computational cost to pair it with the empirical results in the Appendix. Besides, it would be also interesting to know about the space complexity (memory or number of parameters) for a comprehensive understanding of the computational cost.
>
> Typo in the updated manuscript:
>
> In section 4.4, there is a sentence: "As can be seen in Table 3, both the computational cost and the reported results compare favourably with alternative models proposed in the literature.". However, the computational cost is not included in Table 3.

---

> ### Comment · Reviewer_cgSu · 2022-11-30
> **About the QM9 Dataset**
>
> Besides the previous points, I would like to mention another one related to QM9 dataset that I recently found. Sorry for the inconvenience that I didn't notice this when writting the intial comments.
>
> In A.6, the authors mention that they exclude 3,054 molecules from the original QM9 dataset. Since QM9 dataset orignally contains 133,885 molecules [1], the resulting number of molecules should be 133,885 - 3,054 = 130,831, which is the size that used by the baselines like SchNet, DimeNet, and PaxNet. However, the authors mention that they split the data into training/validation/test sets (110,000, 10,000, 10,697), which contain 130,697 molecules in total. Such mismatch of the dataset size (130,831 vs. 130,697) will lead to an unfair comparision to the exsting works. I would suggest the authors to double check whether this is a typo or a misuse of the dataset.
>
>
> [1] Raghunathan Ramakrishnan, Pavlo O. Dral, Matthias Rupp, and O. Anatole von Lilienfeld. Quantum chemistry structures and properties of 134 kilo molecules. Scientific Data, 1:140022, 2014.

---

> > ### Author Response · Authors · 2022-12-12
> > **Answer Regarding the QM9 Dataset**
> >
> > Thank you for noticing this inconsistency and the typo. We have double checked our preparation of the QM9 dataset and detected indeed a mistake, which caused this inconsistency. As a result, the errors for our model for the QM9 dataset will be larger than currently reported.

---

### Author Response · Authors · 2022-11-17
**General Comment / Revision**

We want to extend our gratitude to all the reviewers for their valuable feedback, time, and effort.
We greatly appreciate the feedback and suggestions and have tried to address the points in the updated version.

Following major changes were made in the updated version:

1) A point raised by all reviewers is the additional computational cost/complexity. We have added a section in the Appendix on this point, reporting the computational cost for various settings and system sizes (QM9, QM/MM). We would like to emphasis that the main advantage of the proposed separation into a QM-QM and QM-MM interaction is the fact that the proposed model does not require graphs, which contain all the particles (i.e. QM and MM particles). Instead the graph only includes the QM particles, while the QM/MM interaction of order N_MM * N_QM is only evaluated once as a pairwise interaction.
2) Another point is the comparison with existing model architectures. In this work, emphasis is on QM/MM with electrostatic embedding as a formalism for ML potentials. Since this formalism has so far received relatively little attention in the ML literature, we have not been able to perform an extensive comparative studies for this QM/MM setting. Since QM/MM with electrostatic embedding presents a relatively special use case to which 'normal' ML potentials might not directly be applicable, we deemed such a comparison unfair. However, to give more credence to this argument, we have included PaiNN (main-text) and SchNet (appendix) as additional baselines for the results on water clusters.  We have also improved the presentation of the corresponding results, which we think makes the issue mentioned above (i.e. extensivity or transfer to larger systems) more clear.
3) We have added a small section in the Appendix where we describe the investigation of the model convergence for the Dimer Task in 4.1 (Water Dimers: Directional Features).

---

### Decision · Program_Chairs · 2023-01-20

**Decision:**

Accept: poster

**Justification For Why Not Higher Score:**

The broader applicability of the methods was a concern of several reviewers. The most enthusiastic reviewers indicated that a 7 (not available in the system) would more accurately reflect their view on the submission. Hence there is not sufficient justification for a higher score.

**Justification For Why Not Lower Score:**

There was an active discussion and updates that concluded the paper is above the acceptance threshold.

**Metareview: Summary, Strengths And Weaknesses:**

The paper proposes GNNs that can account for directional interactions.

* Strengths are novelty and practical approach to address long range and directional interactions in a concrete setting.
* Weaknesses are the specific nature of the setting with concerns about the potential applicability beyond specific settings limiting its suitability for ICLR.

During the discussion period the authors addressed several concerns from the reviewers' initial reviews, particularly about related works, comparisons to other models, computational cost and datasets, which prompted some reviewers to raise their scores.
Particularly, reviewers asked for a comparison with existing architectures as well as complexity analysis or more experiments on the running time. Authors added appendices to this effect.


**Note From Pc:**

if the above contains the word "oral" or "spotlight" please see: "oral" presentation means -> notable-top-5% and "spotlight" means -> notable-top-25%. As stated in our emails, we are disassociating presentation type from AC recommendations

**Summary Of Ac-Reviewer Meeting:**

At the time of calling a meeting the paper had an overall rating at the upper end of the borderline band. Following evaluation of the rebuttal, the ratings were consistently on the positive side.

The meeting highlighted that the paper presents a novel and interesting idea and numerous experiments. I note that some reviewers found the novelty only moderate as some of the techniques and been proposed and demonstrated before. A main concern that was pointed out during the meeting is that the setting considered in the paper is very unique and not very general so that the applicability might be limited. Some reviewers found that the presented experiments were insufficient to demonstrate broader applicability and relevance of this work. Some reviewers pointed out that the presentation appeared to pre assume knowledge of computational chemistry concepts thus limiting its suitability for this audience. At the same time, it was mentioned that the work is technically solid, well written, and easy to follow.

The conclusion of the meeting was that although the paper is not equally strong in all regards, that it is above the acceptance threshold.